# Masquerade of Polish Society—Psychological Determinants of COVID-19 Precautionary Behaviors

**DOI:** 10.3390/ijerph20010129

**Published:** 2022-12-22

**Authors:** Natalia Maja Józefacka, Robert Podstawski, Małgorzata Barbara Płoszaj, Elżbieta Szpakiewicz, Mateusz Franciszek Kołek, Andrzej Pomianowski, Gabriela Kania, Anna Niedźwiecka, Dominika Łabno, Aleksander Michalec, Weronika Paw

**Affiliations:** 1Institute of Psychology, Pedagogical University of Krakow, Podchorążych 2, 30-084 Krakow, Poland; 2Department of Tourism, Recreation and Ecology, University of Warmia and Mazury in Olsztyn, 10-957 Olsztyn, Poland; 3Diplomstudium Humanmedizin, Medizinische Universität Wien, Spitalgasse 23, 1090 Wien, Austria; 4Department of Internal Diseases with Clinic, University of Warmia and Mazury in Olsztyn, 10-719 Olsztyn, Poland; 5Students Scientific Club ControlUP, Institute of Psychology, Pedagogical University of Krakow, Podchorążych 2, 30-084 Krakow, Poland

**Keywords:** COVID-19, Dark Triad, social approval, need for cognitive closure, mask wearing, vaccination, empathy

## Abstract

The risk of contracting COVID-19 was a very specific situation of uncertainty and ambi-guity, and of course, cognitively interesting for psychologists studying the determinants of behaviors of different personality types. In this study, we set our sights on trying to find a correlation between adherence to wearing masks and receiving vaccinations and having certain character traits that we thought might influence preventive behavior or not. We focused on the Dark Triad—psychopathy, Machiavellianism and narcissism—as well as social approval and the need for cognition closure, as these traits have previously been linked to heightened conspiracy mentalities. We recruited 159 subjects in the experiment, including 53 male and 106 female participants over the age of 18 to take part in an online survey investigating personality and COVID-19 information. The results confirmed our hypothesis that age, empathy, the need for social approval and other psychological traits are the factors that differentiates people who wear face masks from those who do not. However, it seems impossible to define one set of features that would predispose people to not wear face masks. In our study, the importance of psychological features differed depending on the category of public places. We discuss possible implications of these findings and provide direction for future research.

## 1. Introduction

A few months ago, there was hope that we were in a final phase of the pandemic and forecasts from health politicians and from well-informed newspapers predicted an end of the pandemic for Spring 2022. Currently, the optimism for a forthcoming end of the pandemic is gone. Europe is again the focus of major epidemic activity and the appearance of a new variant—omicron, first described in South Africa, is filling the headlines of newspapers and fueling the political discussions.

During the first wave of the COVID-19 pandemic, many governments imposed various containment measures and scientists explored their efficacy. Interventions imposed by different governments included isolation, closure of schools, test and trace, quarantine, stay-at-home orders, travel restrictions, closure of non-essential businesses, limiting of social interactions, physical distancing, face mask wearing and many others.

This article attempts to address the question of indicators for reluctance to utilize preventive measures such as vaccination and mask wearing. Looking at the effect of other individual intervention would be highly relevant, but data on the efficacy of individual intervention from field trials are still scarce and beyond the scope of this article.

The current COVID-19 outbreak situation is still fresh, so it is important to understand the factors influencing health-promoting behavior in such challenging times. One way to protect yourself from COVID-19 is to wear masks, which, according to studies, block droplet transmission of the virus [1]. Contrary to people’s belief about the harmfulness of masks, they do not cause changes in the oxygen and carbon dioxide content of the body. In addition, they do not adversely affect respiratory rate and volume [2]. In Polish society, there is a general reluctance to wear masks and a lot of misinformation on the subject.

Information on the development of the coronavirus epidemic evidently caused uncertainty and fear of contamination and disease, suffering and even death. As Doliński claims in his article [3], the pandemic had a huge impact on the well-being of people (in this case, the research group consisted of students). In Poland, there were also completely inconsistent communications from the authorities. First, the citizens found out that the use of masks is pointless, and then that it is necessary to use them. At one point, it was forbidden to enter forests and parks, only to be announced later that it is beneficial to health, and therefore even advisable. Many celebrities have stated on multiple occasions that any restrictions are pointless because the virus is gone or is fake [3]. A CBOS [4] survey release shows that more than half of Poles say that wearing masks for prolonged periods of time is harmful to the body. The declarations of Poles on the extent to which they adhered to the recommendations, which were intended to limit the spread of the virus, were also examined. The results [5] show that each month Polish society followed the safety rules less: avoiding public gatherings, keeping a distance of more than one meter in relations with others outside the home, covering the mouth when sneezing and coughing, staying at home as much as possible, taking care of hygiene outside the home by washing hands more often or sanitizing them and wearing masks outside the home. In May and June 2020, 72% of respondents declared that they wear the mask outside the home in such a way that it covers the mouth and nose. On the other hand, 33% of respondents admitted then that they sometimes put the mask on their chin, pull it off their nose or take it off completely, even when there are other people nearby [5].

A decline in the willingness to follow recommendations can be related both to pandemic fatigue and low levels of public confidence, and to the increasing popularity of denial.

Poles’ concerns about the pandemic have significantly decreased, while at the end of 2020, the number of cases and deaths caused by COVID-19 continued to increase [6]. It may be related to the occurrence of the so-called phase of immunity, i.e., getting used to a stressful situation, which, despite the actions we take, does not change. This decline in anxiety may also be explained by the growing tendency to use denial strategies, i.e., coping with stress by thinking that the danger is overflowing, that it is better to live now than regret later. The fact that we were more afraid of the illness of a loved one or the deterioration of the financial situation of a loved one than of such events concerning ourselves could indicate, on the one hand, a high level of care and empathy, and, on the other hand, a phenomenon known as unrealistic optimism [1]. It is related to the belief that the threat we are aware of does not apply to us and that there is a very small chance that we will be exposed to it while it affects our neighbors or colleagues. This means that we are convinced that certain things are happening “next to us” and that they only affect other, random people.

Past studies have also been able to prove that women are more likely than men to exhibit preventive behavior during the COVID-19 pandemic [7].

In particular, it is important to focus on vaccination uptake decisions, which can help stabilize the pandemic, as well as the stricter requirement to wear protective masks, which can reduce the risk and ease of COVID-19 transmission [8].

Overall, pooled analysis [9] showed a significant reduction in COVID-19 incidence with mask wearing, although heterogeneity between studies was substantial. Controlled trials of mask wearing are difficult to conduct, as separating mask wearing effects in population studies from the impact of other regulations is challenging, and the efficacy of masks depend on mask material and mask fit. The combination of vaccination and mask wearing is potentially synergistic since, thus far, vaccination protects well from disease development (the omicron variant is currently an unknown) but immunity from infection wanes over a few months after vaccination. In comparison, masks interfere with the virus transmission process at a level of a physical barrier independent of coronavirus variant.

COVID-19 virus is mainly detected in air, which is responsible for the spreading of the disease [10]. According to the current research, one of the major routes of transmission of COVID-19 virus is primarily from speaking, coughing or sneezing [11,12,13,14,15,16,17,18].

Vaccination and masks are much less costly to apply than other measures, which are associated with high economic and social costs, but paradoxically, both measures are the target of a vocal opposition by a sizable minority of the society. In parallel with biomedical research, we need more social science research into this opposition to guide political decisions on how to end the pandemic.

The risk of contracting COVID-19 places society in a state of uncertainty and ambiguity, resulting from unclear information disseminated by the media. This situation likely causes some individuals to decide against taking preventive measures. It is interesting from a psychological point of view, to determine whether individual character traits are responsible for the decision to spurn or refuse said measures.

Social approval. Social approval is defined as the tendency to portray oneself in a favorable light in order to gain social acceptance and is characterized by the need to behave in ways promoted by society [19]. Research has shown that social approval is associated with prosocial behavior [20], and that in situations of danger or disaster, it influences a strong concern for others, a sense of identity and cooperation, which affects adherence to social norms [21].

Need for cognition. The opposite of need for cognition is mental closure (closed-mindedness). According to the journal Social Issues, people with mental closure tend to see their own point of view as the only correct one, and for this reason the opinions of others are rejected and considered wrong. Previous studies have demonstrated that mental closure is related to a limited information search, a preference for clear and unambiguous judgments and a simplified cognitive process involved in decision making [22,23]. Additionally, preference for predictability, also popularly known as the need for cognitive closure, is a trait characterized by a desire for a quick solution to an ambiguous situation. A person who prefers predictability simultaneously has a low tolerance for cognitive uncertainty, which is associated with succumbing to stereotypes and looking for patterns of behavior. Kossowska describes people with a high need for cognitive closure as preferring order and making decisions easily [24].

Psychopathology. Psychopathology, also known as the Dark Triad of personality, is composed of three negative person-centered personality traits. It consists of Machiavellianism, psychopathy and narcissism. Machiavellianism is defined as a trait characterized by manipulativeness and cynicism, psychopathy as a trait characterized by impulsiveness and emotional callousness and narcissism as a trait characterized by a sense of entitlement and high self-esteem [25]. Studies have shown that these character traits are negatively associated with adherence to restrictions during the COVID-19 pandemic, due to traits such as impulsivity, low concern for others, striving for power, desire for competition and cynicism [26].

Trait psychopathy and Machiavellianism have previously been linked to heightened conspiracy mentalities in general [27]. Dark traits facilitate manipulative and exploitative lifestyles, and these individuals may be more susceptible to general conspiracy beliefs, which perhaps stems from their cynical views and their own attempts to manipulate others [27]. Furthermore, individuals high in the Dark Triad (psychopathy, narcissism, Machiavellianism) personality traits report increased levels of out-group hostility [28] and in-group favoritism, suggesting they may view their in-groups as superior in a similar way to collective narcissists. It is therefore possible that these additional dark personality traits may also be associated with the conspiracy theories and thus not following the norms and rules, such as wearing masks or vaccinating.

Other traits. We also analyze in our research how such a level of empathy and conformity can moderate the relationship between wearing masks and vaccination.

The topic of the COVID-19 pandemic is still a field for research, and it is especially worth focusing on a deeper understanding of the reasons for people’s attitudes toward the restrictions, which will help manage them wisely to best ensure the safety of society as a whole. In this study, we set our sights on trying to find a psychological trait of people who refuse to compliance precaution behavior.

Most studies related to mask wearing were conducted during the early phase of the pandemic. Despite that, little is understood about how people behave in order to mask willingness in different categories of public places. Previous studies focus on psychological traits of people who wearing the mask or agree of vaccinations [29,30,31], although there were no studies of psychological traits of no-willingness of face cover in different public places.

## 2. Materials and Methods

### 2.1. Data Collection

The study used cross-sequential quantitative design. Participants completed an online survey. It was disseminated through social media using the snowball sampling technique. The survey included information on socio-demographics—about age, gender and place of residence, as well as a question about vaccination and about wearing a mask in selected areas. The following parts included five questionnaires about empathy, social approval, need for cognitive closure, conformity and the Dark Triad.

### 2.2. Sample

The sample included 159 subjects from Poland. Most of the participants self-identified as female (106). Mean age was M = 27.64; SD = 11.54, ranging from 18 to 76. The inclusion criteria included age over 18, and current research was directed to the general population.

### 2.3. Measurements

The instruments used in this study were selected to assess a wide range of psychological variables. Socioeconomic factors (age, gender,) were assessed alongside psychological factors such as empathy, social approval, psychopathy and conformity. Behavioral variables related to pandemic-specific precautionary behaviors (face covering, vaccination) were assessed at time of the questionnaire.

Precaution behaviors:

Mask willingness—For the study, we used a scale created by our team to measure willingness to wear masks at shopping malls, workplaces, public transportation and cultural institutions such as museums, theaters, opera houses, stadiums, etc. In this scale, the subjects were asked to specify whether they wear masks or not.

Vaccination—there was the question if the person is vaccinated against COVID-19 or not.

Psychological factors:Empathy—The Empathy Quotient (EQ-Short) (Skrócona Skala Ilorazu Empatii, SSIE) [32]. It consists of 22 questions to which respondents answer on a 4-point Likert scale (1 = definitely no, 2 = rather no, 3 = rather yes, 4 = definitely yes).Social Approval—The Social Approval Questionnaire (KAS) [33]. It consists of 29 questions, which respondents answer by marking the answer “True” or “False”.Need for cognition—Need for Cognition Scale (Skala Potrzeby Poznawczego Domknięcia, SPPD) [34]. It consists of 18 questions to which respondents answer on a 5-point Likert scale (1 = definitely no, 2 = rather no, 3 = hard to say, 4 = rather yes, 5 = definitely yes). It consists of five subscales: preference for order, preference for predictability, intolerance of ambiguity, closed-mindedness and determination.Psychopathy—The Dirty Dozen (Parszywa Dwunastka, DTDDP) [35]. It consists of 12 questions that respondents answer on a 5-point Likert scale (1 = not at all true, 2 = slightly true, 3 = moderately true, 4 = very true, 5 = extremely true). It consists of three subscales: psychopathy, narcissism and Machiavellianism.Conformity—The Conformity Attitudes Scale (SPK-II) (Skala Postaw Konformistycznych, SPK-II) [36]. It consists of 15 questions that respondents answer on a 6-point Likert scale (1 = strongly disagree, 2 = disagree, 3 = rather disagree, 4 = rather agree, 5 = agree, 6 = completely agree). It consists of three subscales: lack of self-confidence, compliance and passivity.

### 2.4. Statistical Analysis

In order to build predictive models for wearing a face and nose covering in public places, statistical analysis using machine learning techniques was performed. Firstly, the reliability of examined scales and subscales was assessed with Cronbach’s α and McDonald’s ω coefficients. Then, the data were split into training and test datasets, in a ratio of 7:3. The best subsets of predictors for wearing a mask in various public places were chosen with a recursive feature elimination (RFE) wrapper algorithm for classification and regression trees (CART). The maximum number of predictors was limited to 8, in order to avoid overfitting to the training dataset. The performance of predictive models was then assessed on test datasets. The global level of significance was set to α = 0.050.

## 3. Results

The results of the reliability analysis are presented in Table 1. Almost all subscales achieved sufficient reliability, which validated their usage in subsequent analyses.

Next, the number of predictors and the predictors themselves for CART models were chosen, using the RFE wrapper algorithm. The results of RFE analysis are presented in Table 2. Evaluation of all models are presented in Table 3.

The models were built exactly under the formulas predefined in the RFE analysis. They were then graphically displayed as classification trees and their accuracy was tested against the test dataset (Figure 1, Figure 2, Figure 3 and Figure 4). In addition, variable importance is presented in Figure 5.

## 4. Discussion

The study examined psychological factors associated with non-compliance precaution practices in a Polish sample during the pandemic period. This article was designed to focus on a deeper understanding of the psychological determinants of wearing face masks as a precautionary behavior. Here, we examined the implicit association between mask wearing and such psychological traits as empathy, social approval, need for closure and conformity. Moreover, as a proliferation of conspiracy theories emerged during the pandemic [37], we focused on the Dark Traid: psychopathy, Machiavellianism and narcissism, which have previously been linked to heightened conspiracy mentalities [27,38].

The results confirmed our hypothesis that age, empathy, the social approval and other psychological traits are the factors that differentiate people who denied compliance to face mask wearing. In our study, we decided to check how psychological characteristics could change willingness to face cover depending on the category of public places. Based on observation, we identified four places where face covering was compulsory. According to the results, there were different traits of mask wearing refusal in each of the recognized places.

In the shopping gallery the most important factors were as follows: age (under 21), empathy (not very high), closed-mindedness and Machiavellianism. Whereas age and empathy seem obvious, Machiavellianism and closed-mindedness require an explanation. Individuals high in Machiavellianism most likely refuse wearing a face mask. If they are additionally closed-minded, then it is even more likely. This personal trait configuration covers 85% of participants who refuse using face cover. These results are in line with previous research that indicated Machiavellianism emerged as a positive predictor of COVID-19 conspiracy beliefs [38], and closed-minded people tend to not question their ideas. Usually, they choose to ignore information that is inconsistent with existing beliefs [39]. Therefore, if they believe that COVID-19 does not exist, they see no reason to wear a face mask.

Main factors for participants who refuse to face covering in the cultural institutions (such as museums, theaters, opera house, etc.) were as follows: age, social approval, need for predictability and preference for order (both are the part of the broader psychological construct: need for cognition) [40]. Individuals aged 24 or younger, with a low need for predictability and not a very high need for social approval, are the people least likely to cover their faces while visiting the cultural institutions. Those young people have no tendency to portray themselves in a favorable light in order to gain social acceptance, do not care about the opinion of others and also do not feel discomfort when confronted with ambiguity. Therefore, they have no external motivation to wear a face mask.

Similarly, in the work environment, age and the need for social approval seem to be crucial for precautionary behavior during the pandemic. Moreover, the psychopathy and narcissism (as the parts of Dark Triad; [41]) and lack of self-confidence are also relevant. Ninety-five percent of narcissistic, but non-psychopathic people who additionally have high lack of self-confidence, do not wear face masks. As narcissism is characterized by an unrealistically positive self-view, feelings of entitlement and a lack of regard for others [42], there is no question why narcissism matters when it comes to not wearing face masks.

On the issue of public transport, making a decision to wear face masks is much more simple than in different places. The two factors that matter are getting vaccinated against COVID-19 and age. When a person is not vaccinated and is older than 20, there is very strong probability that they will not wear a face mask in a public transport. Consistently, when a person is vaccinated, they will wear the face mask. This precaution behavior will be even more supported if the person is empathetic and open-minded.

Previous research has focused on psychological factors characterizing people who are compliant with the COVID-19 restrictions. For example, Woodcock and Schultz [43] suggest that an individual’s decision to engage in behaviors that can help slow the spread of a virus has an important social component. However, to plan actions that result in reducing the spread of virus, it is worth looking at the situation from a broader perspective. Focusing on protecting oneself and other seems insufficient. People who do not compliance the restrictions constitute a wide and heterogeneous group. Their choice is deter-mined not only by psychological factors but also by the category of a public place. Our results suggest, inter alia, that participants were more likely to refuse to wear a mask in cultural institutions and at work. Therefore, our findings have important implications for efforts to promote health protective behaviors. Both in future research and in the information campaigns, it would be worth consider both psychological and social factors. Moreover, expanding research in this area may help shape the behavior of precaution behaviors skeptics It could be helpful in the hypothetical future pandemic situations.

### Limitations and Further Research Directions

This study involved a whole host of different psychological constructs that turned out to be important for refuse to face cover during the COVID-19 pandemic. Although this project can be treated only as a preliminary research report, it can already be seen that the results of the study may play an important role in planning ways to encourage health-related behavior in a similar situation in the future. The snowball sampling technique may be biased by over-representing the academic community with a disproportionate number of highly educated individuals and participants identifying as female. The model lacks the role played by situational variables (i.e., peer pressure, access to information, cultural values and practices), which should be examined in future models. Future research should take into account more variables to create a holistic picture of the reasons why face masks are not worn during the pandemic.

## 5. Conclusions

The current study investigated the psychological factors associated with not wearing face masks in the Polish sample during the pandemic. The results suggest that when discussing reasons for not wearing them, it is not enough to invoke the “General Good” argument. Much depends on the individual psychological characteristics of a particular person. The factors such as age, empathy, closed-mindedness, the Dark Triad and self-confidence seem to be particularly influential. The most interesting result of our study is that these characteristics have different effects depending on the location. While the age of the particular person played an important role in all places, in the shopping malls, traits such as empathy, mental closedness and Machiavellianism were also significant factors in favor of not wearing masks. Considering the cultural places, need for social approval, need for predictability and preference for order seemed important. In the workplaces, on the other hand, social approval, psychopathy, narcissism and self-confidence mattered. Finally, in the public transport, the most important factors were mental openness, empathy and vaccination.

The different configuration of traits and places seems to be important from a practical point of view and may have a different impact on compliance with the restrictions. Therefore, when implementing preventive measures during a pandemic, it is worth paying attention to how the means of communication are used to raise awareness and inform people, so that regardless of individual differences, we will be able to collectively take care of the health of the public. The results of our study may be also helpful in the hypothetical future pandemic situation. They suggest that future information campaigns should be very extensive and take into account many factors to reach as many people as possible.

## Figures and Tables

**Figure 1 ijerph-20-00129-f001:**
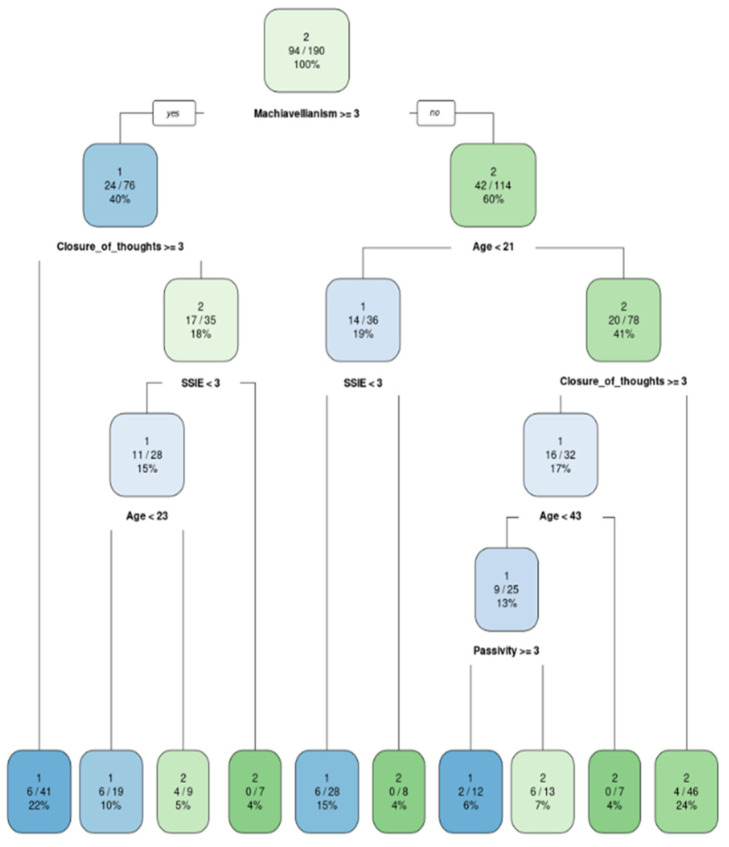
Classification model for a stay in the shopping gallery: 1—wearing a mask; 2—not wearing a mask.

**Figure 2 ijerph-20-00129-f002:**
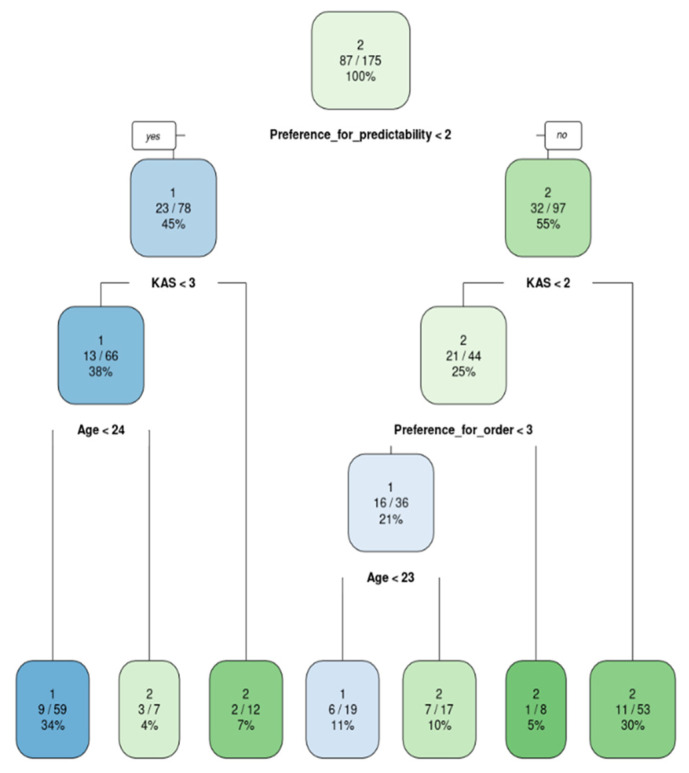
Classification model for a stay in the cultural facilities: 1—wearing a mask; 2—not wearing a mask.

**Figure 3 ijerph-20-00129-f003:**
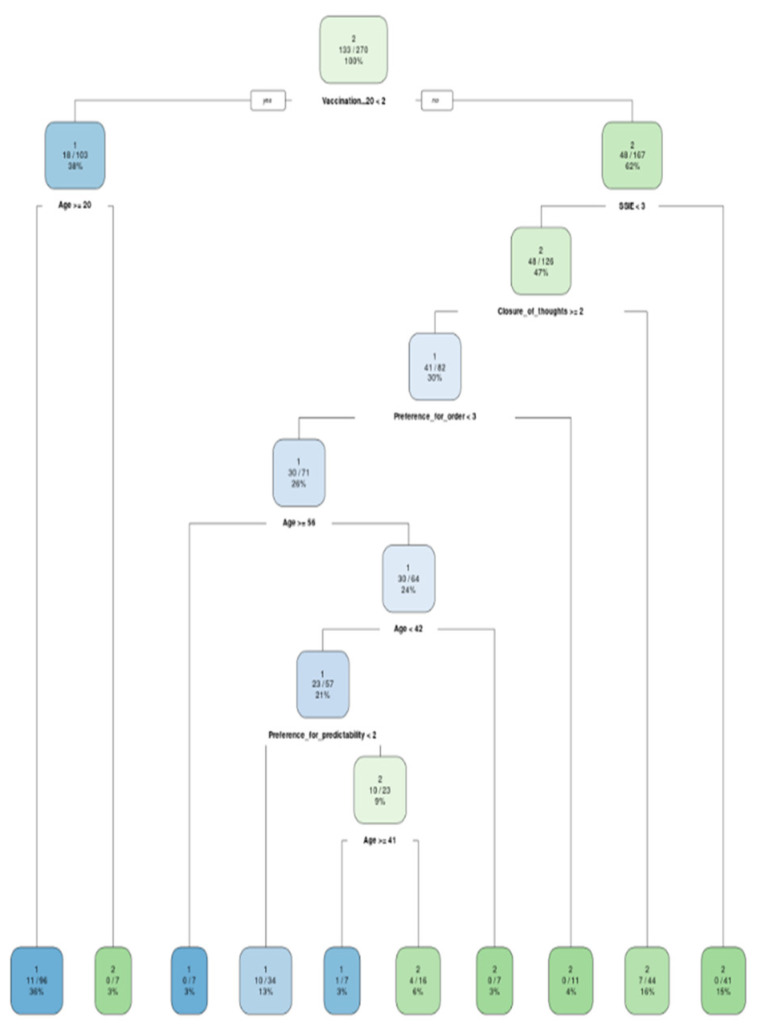
Classification model for a stay in the public transportation: 1—wearing a mask; 2—not wearing a mask.

**Figure 4 ijerph-20-00129-f004:**
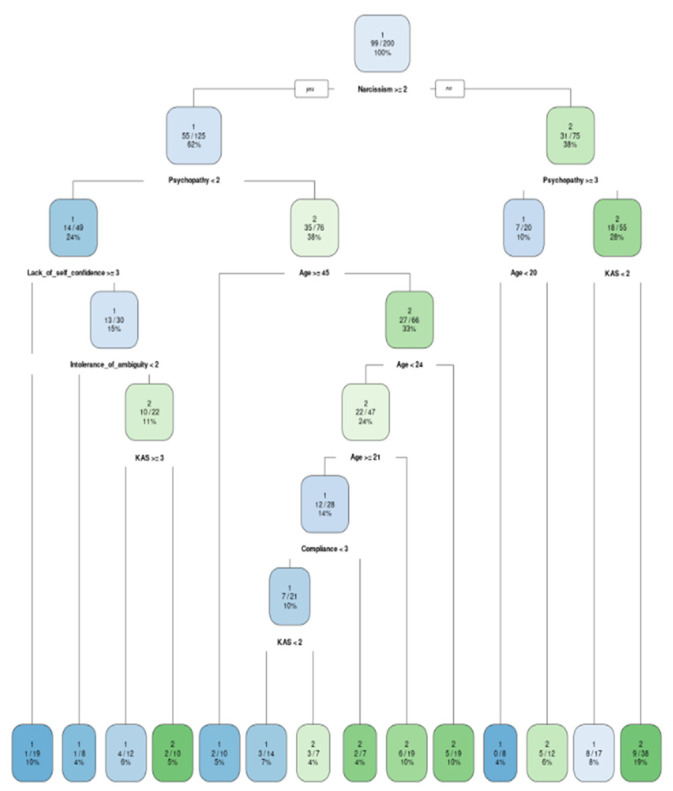
Classification model for a stay in the work: 1—wearing a mask; 2—not wearing a mask.

**Figure 5 ijerph-20-00129-f005:**
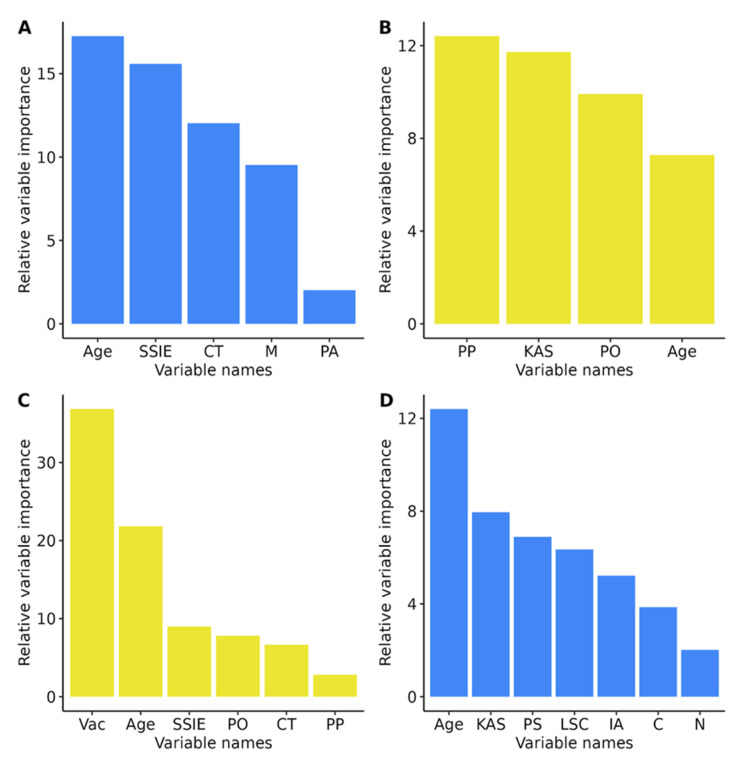
Variable importance for all predictive models: (**A**)—shopping gallery, (**B**)—cultural facility, (**C**)—public transportation, and (**D**)—work. Abbreviations: PP—preference for predictability, LSC—lack of self-confidence, Vac—vaccination status, N—narcissism, D—determination, IA—intolerance of ambiguity, PS—Psychopathy, PA—passivity, C—compliance, CT—closed-mindedness, M—Machiavellianism and PO—preference for order.

**Table 1 ijerph-20-00129-t001:** Reliability analysis.

Variable	α	Omega ω	Mean *r*
SSIE	0.82 (0.78–0.86)	0.83 (0.79–0.92)	0.18
KAS	0.75 (0.69–0.80)	0.75 (0.68–0.88)	0.09
Lack of self-confidence	0.80 (0.75–0.85)	0.82 (0.75–0.96)	0.40
Compliance	0.79 (0.74–0.84)	0.80 (0.72–0.84)	0.36
Passivity	0.76 (0.70–0.82)	0.76 (0.71–0.82)	0.39
SPKII	0.89 (0.86–0.91)	0.91 (0.83–0.98)	0.31
Psychopathy	0.58 (0.46–0.67)	0.63 (0.54–0.79)	0.27
Narcissism	0.82 (0.77–0.86)	0.83 (0.80–0.93)	0.53
Machiavellianism	0.78 (0.72–0.83)	0.79 (0.69–0.93)	0.48
DTDDP	0.81 (0.76–0.85)	0.87 (0.74–0.97)	0.27
Preference for order	0.81 (0.76–0.86)	0.82 (0.63–0.87)	0.60
Preference for predictability	0.66 (0.56–0.74)	0.68 (0.57–0.86)	0.40
Intolerance of ambiguity	0.66 (0.55–0.74)	0.66 (0.62–0.85)	0.39
Closed-mindedness	0.68 (0.58–0.75)	0.70 (0.56–0.89)	0.41
Determination	0.72 (0.64–0.79)	0.74 (0.62–0.88)	0.46
SPPD	0.63 (0.54–0.71)	0.88 (0.71–1.00)	0.10

**Table 2 ijerph-20-00129-t002:** Recursive feature elimination for four distinct predictive models.

Number of Variables	Shopping Gallery		Cultural Institutions		Public Transport		Work	
	Accuracy	Kappa	Accuracy	Kappa	Accuracy	Kappa	Accuracy	Kappa
1	68.57	0.73	59.93	0.20	65.73	0.31	56.51	0.13
2	75.27	0.51	68.02	0.36	79.17	0.58	58.16	0.16
3	83.84	0.68	73.74	0.47	84.10	0.68	60.00	0.20
4	87.70	0.75	76.61	0.53	87.10	0.74	64.52	0.29
5	91.90	0.84	78.62	0.57	89.40	0.79	67.40	0.35
6	90.04	0.80	78.62	0.57	90.62	0.81	68.10	0.36
7	91.06	0.82	80.73	0.62	90.46	0.81	70.03	0.40
8	90.06	0.80	81.65	0.69	92.01	0.84	69.00	0.50

**Table 3 ijerph-20-00129-t003:** Evaluation of built predictive models on training and test datasets.

Statistic	Shopping Gallery		Cultural Institutions		Public Transport		Work	
	Train	Test	Train	Test	Train	Test	Train	Test
Accuracy *	82.11	73.42	77.71	63.51	87.83	87.65	75.89	71.19
Kappa	0.64	0.47	0.55	0.27	0.76	0.75	0.52	0.43
Sensitivity *	85.11	79.49	72.41	56.76	90.80	93.48	70.42	63.33
Specificity *	79.17	67.50	82.95	70.27	85.29	80.00	81.43	79.31
PPV *	80.00	70.45	80.77	65.62	84.04	86.00	79.37	76.00
NPV *	84.44	77.14	75.26	61.90	91.58	90.32	73.08	67.65

Note: *—expressed as percentages.

## Data Availability

Not applicable.

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
