# Peer review of "Masquerade of Polish Society—Psychological Determinants of COVID-19 Precautionary Behaviors"

_ijerph, 2022, doi:10.3390/ijerph20010129_

Round 1

Reviewer 1 Report

The authors present interesting results that may provide important elements to consider in the adoption of measures in specific contexts to prevent new outbreaks by covid-19.

However, it leaves some doubts in terms of methodology.

1) refer data was split into training and test datasets, in a ratio of 7:3. Why this ratio?

2) The maximum number of predictors was limited to 8. I understand the procedure, but it is not clear which variables were integrated and which were excluded.

3) The data presented in the tree diagram (figures 1-4) is not self-explanatory (or even visible, given its small size), which makes it difficult to understand the results. I think that using the optimal tree diagram would have been a better option.

4) the choice of sample (by convenience) can introduce several biases in the results. What procedures were introduced to control this effect?

The results lacked further discussion.

More discussion of the results was lacking. The conclusions are limited in relation to the results, and make little contribution to the implementation of preventive measures.

It also presents several typos and construction errors, especially in the literature review. A revision of the English language is suggested.

Author Response

Dear Reviewer,

Enclosed is our response to the review.

Best regards,

Robert Podstawski

Reviewer 2 Report

Thank you for giving me the opportunity to review this manuscript. I want to congratulate the authors for the great work that has been put into the paper. However, I have some  comments :

The method section needs more elaboration on the following section 

the exclusion and inclusion criteria of the participants

the procedure of recruitment and data collection

ethical consideration? 

Also, the measurement section needs more specifications, such as the reliability and validity scores of the tool, etc. 

 Finally, where are the implications of the study? 

Author Response

(The authors gave the same response as above.)

Reviewer 3 Report

Mask-wearing, social distance and hygiene rules have been the most important protective measures that have been emphasized since the covid-19 pandemic started, but in many countries, problems have been encountered with the society's compliance with these rules. This study is important in terms of revealing the obstacles to complying with the rules that reduce the risk of contamination.

minor revision is required.

1-There is a grammatical error on page 3, line 121. "so far" words exlude from the sentence.

2-The sentence  begin with “This article was designed to focus on a deeper understanding of the psychological determinants of wearing  face masks ………...” on page 11, line 285 is written twice, remove one of them.

Author Response

(The authors gave the same response as above.)

Round 2

Reviewer 1 Report

The authors followed the reviewers' recommendations, providing necessary additional elements (methodology) and further discussion of the results.

Reviewer 2 Report

Thank you for addressing my comments